# Communication Efficient Distributed Machine Learning with the Parameter Server

**Mu Li**[*†]**, David G. Andersen**[*]**, Alexander Smola**[*‡]**, and Kai Yu**[†]
[*]Carnegie Mellon University  [†]Baidu  [‡]Google
{muli, dga}@cs.cmu.edu, alex@smola.org, yukai@baidu.com

## Abstract

This paper describes a third-generation parameter server framework for distributed machine learning. This framework offers two relaxations to balance system performance and algorithm efficiency. We propose a new algorithm that takes advantage of this framework to solve non-convex non-smooth problems with convergence guarantees. We present an in-depth analysis of two large scale machine learning problems ranging from $\ell_1$-regularized logistic regression on CPUs to reconstruction ICA on GPUs, using 636TB of real data with hundreds of billions of samples and dimensions. We demonstrate using these examples that the parameter server framework is an effective and straightforward way to scale machine learning to larger problems and systems than have been previously achieved.

## 1 Introduction

In realistic industrial machine learning applications the datasets range from 1TB to 1PB. For example, a social network with 100 million users and 1KB data per user has 100TB. Problems in online advertising and user generated content analysis have complexities of similar order of magnitudes [12]. Such huge quantities of data allow learning powerful and complex models with $10^9$ to $10^{12}$ parameters [9], at which scale a single machine is often not powerful enough to complete these tasks in time.

Distributed optimization is becoming a key tool for solving large scale machine learning problems [1, 3, 10, 21, 19]. The workloads are partitioned into worker machines, which access the globally shared model as they simultaneously perform local computations to refine the model. However, efficient implementations of the distributed optimization algorithms for machine learning applications are not easy. A major challenge is the inter-machine data communication:

- Worker machines must frequently read and write the global shared parameters. This massive data access requires an enormous amount of network bandwidth. However, bandwidth is one of the scarcest resources in datacenters [6], often 10-100 times smaller than memory bandwidth and shared among all running applications and machines. This leads to a huge communication overhead and becomes a bottleneck for distributed optimization algorithms.
- Many optimization algorithms are sequential, requiring frequent synchronization among worker machines. In each synchronization, all machines need to wait the slowest machine. However, due to imperfect workload partition, network congestion, or interference by other running jobs, slow machines are inevitable, which then becomes another bottleneck.

In this work, we build upon our prior work designing an open-source third generation parameter server framework [4] to understand the scope of machine learning algorithms to which it can be applied, and to what benefit. Figure 1 gives an overview of the scale of the largest machine learning experiments performed on a number of state-of-the-art systems. We confirmed with the authors of these systems whenever possible.

Compared to these systems, our parameter server is several orders of magnitude more scalable in terms of both parameters and nodes. The parameter server communicates data asynchronously to reduce the communication cost. The resulting data inconsistency is a trade-off between the system performance and the algorithm convergence rate. The system offers two relaxations to address data (in)consistency: First, rather than arguing for a specific consistency model [29, 7, 15], we support flexible consistency models. Second, the system allows user-specific filters for fine-grained consistency management. Besides, the system provides other features such as data replication, instantaneous failover, and elastic scalability.

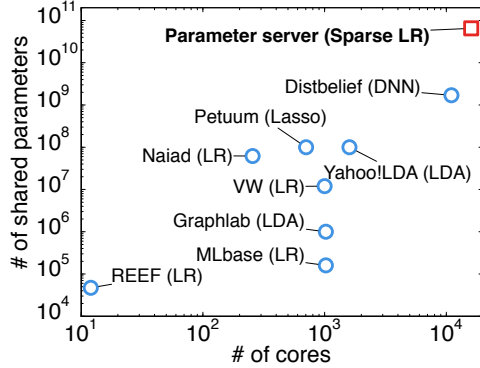

Figure 1: Comparison of the public largest machine learning experiments each system performed. The results are current as of April 2014.

**Motivating Application.** Consider the following general regularized optimization problem:

$$\underset{w}{\text{minimize}} \ F(w) \ \text{where} \ F(w) := f(w) + h(w) \ \text{and} \ w \in \mathbb{R}^p, \tag{1}$$

We assume that the loss function $f : \mathbb{R}^p \to \mathbb{R}$ is continuously differentiable but not necessarily convex, and the regularizer $h : \mathbb{R}^p \to \mathbb{R}$ is convex, left side continuous, block separable, but possibly non-smooth.

The proposed algorithm solves this problem based on the proximal gradient method [23]. However, it differs with the later in four aspects to efficiently tackle very high dimensional and sparse data:

- Only a subset (block) of coordinates is updated in each time: (block) Gauss-Seidel updates are shown to be efficient on sparse data [36, 27].
- The model a worker maintains is only partially consistent with other machines, due to asynchronous data communication.
- The proximal operator uses coordinate-specific learning rates to adapt progress to sparsity pattern inherent in the data.
- Only coordinates that would change the associated model weights are communicated to reduce network traffic.

We demonstrate the efficiency of the proposed algorithm by applying it to two challenging problems: (1) non-smooth $\ell_1$-regularized logistic regression on sparse text datasets with over 100 billion examples and features; (2) a non-convex and non-smooth ICA reconstruction problem [18], extracting billions of sparse features from dense image data. We show that the combination of the proposed algorithm and system effectively reduces both the communication cost and programming effort. In particular, 300 lines of codes suffice to implement $\ell_1$-regularized logistic regression with nearly no communication overhead for industrial-scale problems.

**Outline:** We first provide background in Section 2. Next, we address the two relaxations in Section 3 and the proposed algorithm in Section 4. In Section 5 (and also Appendix B and C), we present the applications with the experimental results. We conclude with a discussion in Section 6.

## 2 Background

**Related Work.** The parameter server framework [29] has proliferated both in academia and in industry. Related systems have been implemented at Amazon, Baidu, Facebook, Google [10], Microsoft, and Yahoo [2]. There are also open source codes, such as YahooLDA [2] and Petuum [15].

As introduced in [29, 2], the first generation of the parameter servers lacked flexibility and performance. The second generation parameter servers were application specific, exemplified by Distbelief [10] and the synchronization mechanism in [20]. Petuum modified YahooLDA by imposing bounded delay instead of eventual consistency and aimed for a general platform [15], but it placed

more constraints on the threading model of worker machines. Compared to previous work, our third generation system greatly improves system performance, and also provides flexibility and fault tolerance.

Beyond the parameter server, there exist many general-purpose distributed systems for machine learning applications. Many mandate synchronous and iterative communication. For example, Mahout [5], based on Hadoop [13] and MLI [30], based on Spark [37], both adopt the iterative MapReduce framework [11]. On the other hand, Graphlab [21] supports global parameter synchronization on a best effort basis. These systems scale well to few hundreds of nodes, primarily on dedicated research clusters. However, at a larger scale the synchronization requirement creates performance bottlenecks. The primary advantage over these systems is the flexibility of consistency models offered by the parameter server.

There is also a growing interest in asynchronous algorithms. Shotgun [7], as a part of Graphlab, performs parallel coordinate descent for solving $\ell_1$ optimization problems. Other methods partition observations over several machines and update the model in a data parallel fashion [34, 17, 38, 3, 1, 19]. Lock-free variants were proposed in Hogwild [26]. Mixed variants which partition data and parameters into non-overlapping components were introduced in [33], albeit at the price of having to move or replicate data on several machines. Lastly, the NIPS framework [31] discusses general non-convex approximate proximal methods.

The proposed algorithm differs from existing approaches mainly in two aspects. First, we focus on solving large scale problems. Given the size of data and the limited network bandwidth, neither the shared memory approach of Shotgun and Hogwild nor moving the entire data during training is desirable. Second, we aim at solving general non-convex and non-smooth composite objective functions. Different to [31], we derive a convergence theorem with weaker assumptions, and furthermore we carry out experiments that are of many orders of magnitude larger scale.

**The Parameter Server Architecture.** An instance of the parameter server [4] contains a server group and several worker groups, in which a group has several machines. Each machine in the server group maintains a portion of the global parameters, and all servers communicate with each other to replicate and/or migrate parameters for reliability and scaling.

A worker stores only a portion of the training data and it computes the local gradients or other statistics. Workers communicate only with the servers to retrieve and update the shared parameters. In each worker group, there might be a scheduler machine, which assigns workloads to workers as well as monitors their progress. When workers are added or removed from the group, the scheduler can reschedule the unfinished workloads. Each worker group runs an application, thus allowing for multi-tenancy. For example, an ad-serving system and an inference algorithm can run concurrently in different worker groups.

The shared model parameters are represented as sorted (key,value) pairs. Alternatively we can view this as a sparse vector or matrix that interacts with the training data through the built-in multi-threaded linear algebra functions. Data exchange can be achieved via two operations: `push` and `pull`. A worker can push all (key, value) pairs within a range to servers, or pull the corresponding values from the servers.

**Distributed Subgradient Descent.** For the motivating example introduced in (1), we can implement a standard distributed subgradient descent algorithm [34] using the parameter server. As illustrated in Figure 2 and Algorithm 1, training data is partitioned and distributed among all the workers. The model $w$ is learned iteratively. In each iteration, each worker computes the local gradients using its own training data, and the servers aggregate these gradients to update the globally shared parameter $w$. Then the workers retrieve the updated weights from the servers.

A worker needs the model $w$ to compute the gradients. However, for very high-dimensional training data, the model may not fit in a worker. Fortunately, such data are often sparse, and a worker typically only requires a subset of the model. To illustrate this point, we randomly assigned samples in the dataset used in Section 5 to workers, and then counted the model parameters a worker needed for computing gradients. We found that when using 100 workers, the average worker only needs 7.8% of the model. With 10,000 workers this reduces to 0.15%. Therefore, despite the large total size of $w$, the working set of $w$ needed by a particular worker can be cached trivially.

**Algorithm 1** Distributed Subgradient Descent Solving (1) in the Parameter Server

---

**Worker $r = 1, \ldots, m$:**

1: Load a part of training data $\{y_{i_k}, x_{i_k}\}_{k=1}^{n_r}$
2: Pull the working set $w_r^{(0)}$ from servers
3: **for** $t = 1$ **to** $T$ **do**
4:     Gradient $g_r^{(t)} \leftarrow \sum_{k=1}^{n_r} \partial\ell(x_{i_k}, y_{i_k}, w_r^{(t)})$
5:     Push $g_r^{(t)}$ to servers
6:     Pull $w_r^{(t+1)}$ from servers
7: **end for**

**Servers:**

1: **for** $t = 1$ **to** $T$ **do**
2:     Aggregate $g^{(t)} \leftarrow \sum_{r=1}^{m} g_r^{(t)}$
3:     $w^{(t+1)} \leftarrow w^{(t)} - \eta\left(g^{(t)} + \partial h(w^{(t)})\right)$
4: **end for**

---

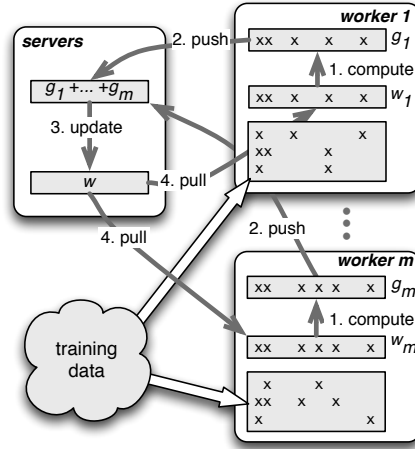

Figure 2: One iteration of Algorithm 1. Each worker only caches the working set of $w$.

## 3 Two Relaxations of Data Consistency

We now introduce the two relaxations that are key to the proposed system. We encourage the reader interested in systems details such as server key layout, elastic scalability, and continuous fault tolerance, to see our prior work [4].

### 3.1 Asynchronous Task Dependency

We decompose the workloads in the parameter server into tasks that are issued by a caller to a remote callee. There is considerable flexibility in terms of what constitutes a task: for instance, a task can be a push or a pull that a worker issues to servers, or a user-defined function that the scheduler issues to any node, such as an iteration in the distributed subgradient algorithm. Tasks can also contains subtasks. For example, a worker performs one push and one pull per iteration in Algorithm 1.

Tasks are executed asynchronously: the caller can perform further computation immediately after issuing a task. The caller marks a task as finished only once it receives the callee's reply. A reply could be the function return of a user-defined function, the (key,value) pairs requested by the pull, or an empty acknowledgement. The callee marks a task as finished only if the call of the task is returned and all subtasks issued by this call are finished.

By default callees execute tasks in parallel for best performance. A caller wishing to render task execution sequential can insert an *execute-after-finished* dependency between tasks. The diagram on the right illustrates the execution of three tasks. Tasks

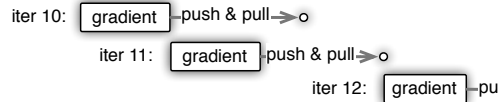

10 and 11 are independent, but 12 depends on 11. The callee therefore begins task 11 immediately after the gradients are computed in task 10. Task 12, however, is postponed to after pull of 11.

Task dependencies aid implementing algorithm logic. For example, the aggregation logic at servers in Algorithm 1 can be implemented by having the updating task depend on the push tasks of all workers. In this way, the weight $w$ is updated only after all worker gradients have been aggregated.

### 3.2 Flexible Consistency Models via Task Dependency Graphs

The dependency graph introduced above can be used to relax consistency requirements. Independent tasks improve the system efficiency by parallelizing the usage of CPU, disk and network bandwidth. However, this may lead to data inconsistency between nodes. In the diagram above, the worker $r$ starts iteration 11 before the updated model $w_r^{(11)}$ is pulled back, thus it uses the outdated model $w_r^{(10)}$ and compute the same gradient as it did in iteration 10, namely $g_r^{(11)} = g_r^{(10)}$. This inconsis-

tency can potentially slows down the convergence speed of Algorithm 1. However, some algorithms may be less sensitive to this inconsistency. For example, if only a block of $w$ is updated in each iteration of Algorithm 2, starting iteration 11 without waiting for 10 causes only a portion of $w$ to be inconsistent.

The trade-off between algorithm efficiency and system performance depends on various factors in practice, such as feature correlation, hardware capacity, datacenter load, etc. Unlike other systems that force the algorithm designer to adopt a specific consistency model that may be ill-suited to the real situations, the parameter server can provide full flexibility for different consistency models by creating task dependency graphs, which are directed acyclic graphs defined by tasks with their dependencies. Consider the following three examples:

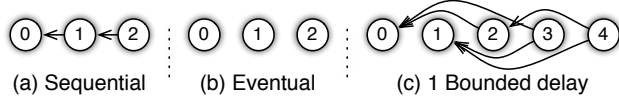

(a) Sequential    (b) Eventual    (c) 1 Bounded delay

**Sequential Consistency** requires all tasks to be executed one by one. The next task can be started only if the previous one has finished. It produces results identical to the single-thread implementation. Bulk Synchronous Processing uses this approach.

**Eventual Consistency** to the contrary allows all tasks to be started simultaneously. [29] describe such a system for LDA. This approach is only recommendable whenever the underlying algorithms are very robust with regard to delays.

**Bounded Delay** limits the staleness of parameters. When a maximal delay time $\tau$ is set, a new task will be blocked until all previous tasks $\tau$ times ago have been finished ($\tau = 0$ yields sequential consistency and for $\tau = \infty$ we recover eventual consistency). Algorithm 2 uses such a model.

Note that dependency graphs allow for more advanced consistency models. For example, the scheduler may increase or decrease the maximal delay according to the runtime progress to dynamically balance the efficiency-convergence trade-off.

## 3.3 Flexible Consistency Models via User-defined Filters

Task dependency graphs manage data consistency between tasks. User-defined filters allow for a more fine-grained control of consistency (e.g. within a task). A filter can transform and selectively synchronize the the (key,value) pairs communicated in a task. Several filters can be applied together for better data compression. Some example filters are:

**Significantly modified filter:** it only pushes entries that have changed by more than a threshold since synchronized last time.

**Random skip filter:** it subsamples entries before sending. They are skipped in calculations.

**KKT filter:** it takes advantage of the optimality condition when solving the proximal operator: a worker only pushes gradients that are likely to affect the weights on the servers. We will discuss it in more detail in section 5.

**Key caching filter:** Each time a range of (key,value) pairs is communicated because of the range-based `push` and `pull`. When the same range is chosen again, it is likely that only values are modified while the keys are unchanged. If both the sender and receiver have cached these keys, the sender then only needs to send the values with a signature of the keys. Therefore, we effectively double the network bandwidth.

**Compressing filter:** The values communicated are often compressible numbers, such as zeros, small integers, and floating point numbers with more than enough precision. This filter reduces the data size by using lossless or lossy data compression algorithms[1].

## 4   Delayed Block Proximal Gradient Method

In this section, we propose an efficient algorithm taking advantage of the parameter server to solve the previously defined nonconvex and nonsmooth optimization problem (1).

**Algorithm 2** Delayed Block Proximal Gradient Method Solving (1)
___
**Scheduler:**
 1: Partition parameters into $k$ blocks $b_1, \ldots, b_k$
 2: **for** $t = 1$ **to** $T$**:** Pick a block $b_{i_t}$ and issue the task to workers
**Worker $r$ at iteration $t$**
 1: Wait until all iterations before $t - \tau$ are finished
 2: Compute first-order gradient $g_r^{(t)}$ and coordinate-specific learning rates $u_r^{(t)}$ on block $b_{i_t}$
 3: Push $g_r^{(t)}$ and $u_r^{(t)}$ to servers with user-defined filters, e.g., the random skip or the KKT filter
 4: Pull $w_r^{(t+1)}$ from servers with user-defined filters, e.g., the significantly modified filter
**Servers at iteration $t$**
 1: Aggregate $g^{(t)}$ and $u^{(t)}$
 2: Solve the generalized proximal operator (2) $w^{(t+1)} \leftarrow \mathrm{Prox}_{\gamma_t}^U(w^{(t)})$ with $U = \mathrm{diag}(u^{(t)})$.
___

**Proximal Gradient Methods.** For a closed proper convex function $h(x) : \mathbb{R}^p \rightarrow \mathbb{R} \cup \{\infty\}$ define the generalized proximal operator

$$\mathrm{Prox}_\gamma^U(x) := \underset{y \in \mathbb{R}^p}{\mathrm{argmin}} \; h(y) + \frac{1}{2\gamma} \|x - y\|_U^2 \quad \text{where } \|x\|_U^2 := x^\top U x. \tag{2}$$

The Mahalanobis norm $\|x\|_U$ is taken with respect to a positive semidefinite matrix $U \succeq 0$. Many proximal algorithms choose $U = \mathbf{1}$. To minimize the composite objective function $f(w) + h(w)$, proximal gradient algorithms update $w$ in two steps: a forward step performing steepest gradient descent on $f$ and a backward step carrying out projection using $h$. Given learning rate $\gamma_t > 0$ at iteration $t$ these two steps can be written as

$$w^{(t+1)} = \mathrm{Prox}_{\gamma_t}^U \left[ w^{(t)} - \gamma_t \nabla f(w^{(t)}) \right] \quad \text{for } t = 1, 2, \ldots \tag{3}$$

**Algorithm.** We relax the consistency model of the proximal gradient methods with a block scheme to reduce the sensitivity to data inconsistency. The proposed algorithm is shown in Algorithm 2. It differs from the standard method as well as Algorithm 1 in four substantial ways to take advantage of the opportunities offered by the parameter server and to handle high-dimensional sparse data.

1. Only a block of parameters is updated per iteration.
2. The workers compute both gradients and coordinate-specific learning rates, e.g., the diagonal part of the second derivative, on this block.
3. Iterations are asynchronous. We use a bounded-delay model over iterations.
4. We employ user-defined filters to suppress transmission of parts of data whose effect on the model is likely to be negligible.

**Convergence Analysis.** To prove convergence we need to make a number of assumptions. As before, we decompose the loss $f$ into blocks $f_i$ associated with the training data stored by worker $i$, that is $f = \sum_i f_i$. Next we assume that block $b_t$ is chosen at iteration $t$. A key assumption is that for given parameter changes the rate of change in the gradients of $f$ is bounded. More specifically, we need to bound the change affecting the very block and the amount of "crosstalk" to other blocks.

**Assumption 1 (Block Lipschitz Continuity)** *There exists positive constants $L_{var,i}$ and $L_{cov,i}$ such that for any iteration $t$ and all $x, y \in \mathbb{R}^p$ with $x_i = y_i$ for any $i \notin b_t$ we have*

$$\|\nabla_{b_t} f_i(x) - \nabla_{b_t} f_i(y)\| \leq L_{var,i} \|x - y\| \quad \text{for } 1 \leq i \leq m \tag{4a}$$
$$\|\nabla_{b_s} f_i(x) - \nabla_{b_s} f_i(y)\| \leq L_{cov,i} \|x - y\| \quad \text{for } 1 \leq i \leq m, t < s \leq t + \tau \tag{4b}$$

*where $\nabla_b f(x)$ is the block $b$ of $\nabla f(x)$. Further define $L_{var} := \sum_{i=1}^m L_{var,i}$ and $L_{cov} := \sum_{i=1}^m L_{cov,i}$.*

The following Theorem 2 indicates that this algorithm converges to a stationary point under the relaxed consistency model, provided that a suitable learning rate is chosen. Note that since the overall objective is nonconvex, no guarantees of optimality are possible in general.

**Theorem 2** *Assume that updates are performed with a delay bounded by $\tau$, also assume that we apply a random skip filter on pushing gradients and a significantly-modified filter on pulling weights with threshold $\mathcal{O}(t^{-1})$. Moreover assume that gradients of the loss are Lipschitz continuous as per Assumption 1. Denote by $M_t$ the minimal coordinate-specific learning rate at time t. For any $\epsilon > 0$, Algorithm 2 converges to a stationary point in expectation if the learning rate $\gamma_t$ satisfies*

$$\gamma_t \leq \frac{M_t}{L_{var} + \tau L_{cov} + \epsilon} \quad \text{for all } t > 0. \tag{5}$$

The proof is shown in Appendix A. Intuitively, the difference between $w^{(t-\tau)}$ and $w^{(t)}$ will be small when reaching a stationary point. As a consequence, also the change in gradients will vanish. The inexact gradient obtained by delayed and inexact model, therefore, is likely a good approximation of the true gradient, so the convergence results of proximal gradient methods can be applied.

Note that, when the delay increase, we should decrease the learning rate to guarantee convergence. However, a larger value is possible when careful block partition and order are chosen. For example, if features in a block are less correlated then $L_{\text{var}}$ decreases. If the block is less related to the previous blocks, then $L_{\text{cov}}$ decreases, as also exploited in [26, 7].

## 5 Experiments

We now show how the general framework discussed above can be used to solve challenging machine learning problems. Due to space constraints we only present experimental results for a 0.6PB dataset below. Details on smaller datasets are relegated to Appendix B. Moreover, we discuss non-smooth Reconstruction ICA in Appendix C.

**Setup.** We chose $\ell_1$-regularized logistic regression for evaluation because that it is one of the most popular algorithms used in industry for large scale risk minimization [9]. We collected an ad click prediction dataset with 170 billion samples and 65 billion unique features. The uncompressed dataset size is 636TB. We ran the parameter server on 1000 machines, each with 16 CPU cores, 192GB DRAM, and connected by 10 Gb Ethernet. 800 machines acted as workers, and 200 were servers. The cluster was in concurrent use by other jobs during operation.

**Algorithm.** We adopted Algorithm 2 with upper bounds of the diagonal entries of the Hessian as the coordinate-specific learning rates. Features were randomly split into 580 blocks according the feature group information. We chose a fixed learning rate by observing the convergence speed.

We designed a Karush-Kuhn-Tucker (KKT) filter to skip inactive coordinates. It is analogous to the active-set selection strategies of SVM optimization [16] and active set selectors [22]. Assume $w_k = 0$ for coordinate $k$ and $g_k$ the current gradient. According to the optimality condition of the proximal operator, also known as soft-shrinkage operator, $w_k$ will remain 0 if $|g_k| \leq \lambda$. Therefore, it is not necessary for a worker to send $g_k$ (as well as $u_k$). We use an old value $\hat{g}_k$ to approximate $g_k$ to further avoid computing $g_k$. Thus, coordinate $k$ will be skipped in the KKT filter if $|\hat{g}_k| \leq \lambda - \delta$, where $\delta \in [0, \lambda]$ controls how aggressive the filtering is.

**Implementation.** To the best of our knowledge, no open source system can scale sparse logistic regression to the scale described in this paper. Graphlab provides only a multi-threaded, single machine implementation. We compared it with ours in Appendix B. Mlbase, Petuum and REEF do not support sparse logistic regression (as confirmed with the authors in 4/2014). We compare the parameter server with two special-purpose second general parameter servers, named System A and B, developed by a large Internet company.

Both system A and B adopt the sequential consistency model, but the former uses a variant of L-BFGS while the latter runs a similar algorithm as ours. Notably, both systems consist of more than 10K lines of code. The parameter server only requires 300 lines of code for the same functionality as System B (the latter was developed by an author of this paper). The parameter server successfully moves most of the system complexity from the algorithmic implementation into reusable components.

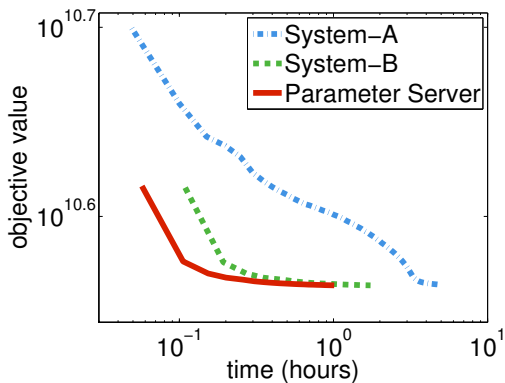

Figure 3: Convergence of sparse logistic regression on a 636TB dataset.

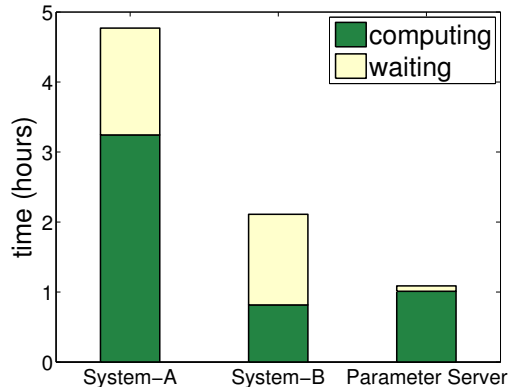

Figure 4: Average time per worker spent on computation and waiting during optimization.

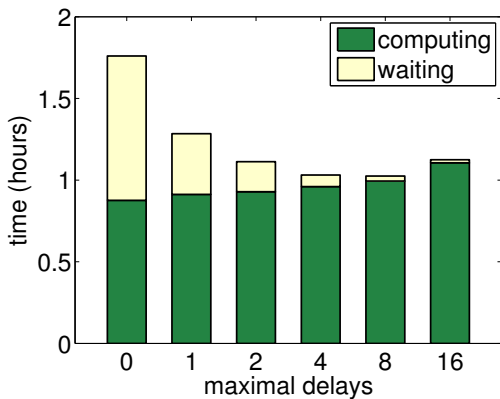

Figure 5: Time to reach the same convergence criteria under various allowed delays.

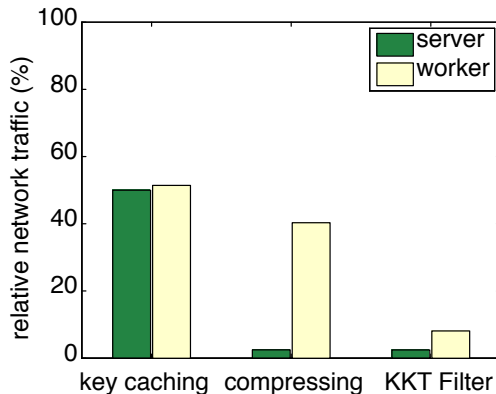

Figure 6: The reduction of sent data size when stacking various filters together.

**Experimental Results.** We compare these systems by running them to reach the same convergence criteria. Figure 3 shows that System B outperforms system A due to its better algorithm. The parameter server, in turn, speeds up System B in 2 times while using essentially the same algorithm. It achieves this because the consistency relaxations significantly reduce the waiting time (Figure 4).

Figure 5 shows that increasing the allowed delays significantly decreases the waiting time though slightly slows the convergence. The best trade-off is 8-delay, which results in a 1.6x speedup comparing the sequential consistency model. As can be seen in Figure 6, key caching saves 50% network traffic. Compressing reduce servers' traffic significantly due to the model sparsity, while it is less effective for workers because the gradients are often non-zeros. But these gradients can be filtered efficiently by the KKT filter. In total, these filters give 40x and 12x compression rates for servers and workers, respectively.

## 6 Conclusion

This paper examined the application of a third-generation parameter server framework to modern distributed machine learning algorithms. We show that it is possible to design algorithms well suited to this framework; in this case, an asynchronous block proximal gradient method to solve general non-convex and non-smooth problems, with provable convergence. This algorithm is a good match to the relaxations available in the parameter server framework: controllable asynchrony via task dependencies and user-definable filters to reduce data communication volumes. We showed experiments for several challenging tasks on real datasets up to 0.6PB size with hundreds billions samples and features to demonstrate its efficiency. We believe that this third-generation parameter server is an important and useful building block for scalable machine learning. Finally, the source codes are available at http://parameterserver.org.

## Footnotes

[1]Both key caching and data compressing are presented as system-level optimization in the prior work [4], here we generalize them into user-defined filters.

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
