[Supplementary Material]

# A Proof of Convergence for Delayed Block Proximal Gradient

For the proof of Theorem 2 we need several technical lemmas. Denote by $b \subseteq \{1, \ldots p\}$ a subset of coordinates and let $x_b \in \mathbb{R}^p$ be the vector obtained by setting the entries of $x$ which are not in block $b$ to 0. We first show that under Assumption 1 the objective is well behaved under subspace shifts.

**Lemma 3** *Assume block $b$ is chosen at time $t$, then the following holds under Assumption 1 for any $f_i$ and for any time $t$, for any $x, y \in \mathbb{R}^p$*

$$f_i(x + y_b) \leq f_i(x) + \langle \nabla f_i(x), y_b \rangle + \frac{L_{var,i}}{2} \|y_b\|^2, \tag{6}$$

**Proof.** By the mean value theorem it follows that

$$f_i(x + y_b) = f_i(x) + \langle \nabla f_i(x + \xi y_b), y_b \rangle \text{ for some } \xi \in [0, 1]. \tag{7}$$

Using the Lipschitz property of Assumption 1 it follows that the gradient at $x + \xi y_b$ can be bounded via $|\nabla f_i(x + \xi y_b) - \nabla f_i(x)| \leq L_{var,i} \xi \|y_b\|$. Combining this with $\xi \leq 1$ proves the claim. ∎

Next we prove that for block separable regularizers the solutions also satisfy an appropriate decomposition property:

**Lemma 4** *Assume that $h$ is block separable and $0 \in \partial h(0)$ and that $U$ is diagonal. For any $x$ and for $\gamma > 0$ we denote by $z = Prox_\gamma^U(x)$ and by $z_b = Prox_\gamma^U(x_b)$ the solutions of the proximal operator to the full vector, and only a subset, respectively. Then for any block $b$ the following holds:*

$$U(x_b - z_b) \in \gamma \partial h(z_b) \tag{8}$$

**Proof.** Since $0 \in \partial h(0)$ it follows that $\text{Prox}_\gamma(0) = 0$. Further since $h$ is block separable, the proximity function $h(y) + \frac{1}{2\gamma} \|x - y\|_U^2$ is also block separable. $z_b = \text{Prox}_\gamma(x_b)$ follows from this by setting all entries of $x$ except those in block $b$ to 0. Finally, (8) follows by taking derivatives on both sides of the definition of proximal operator. ∎

Denote by $\tilde{g}^{(t)}$ and $\tilde{u}^{(t)}$ the aggregated gradients and scaling coefficients at server nodes respectively. Assume that each worker randomly skips a coordinate with probability $1 - q$, where $0 < q < 1$. Let $g^{(t)} := q^{-1}\tilde{g}^{(t)}$ and $u^{(t)} := q^{-1}\tilde{u}^{(t)}$) be the unbiased inexact gradient and scaling coefficient estimates respectively (note that more sophisticated subsampling techniques such as reservoir sampling could be employed, too).

The next step is to bound the changes of the objective function between subsequent iterations $t$ and $t+1$ using the updates $\Delta^{(t)} = w^{(t+1)} - w^{(t)}$ together with the difference between $g^{(t)}$ and $\nabla f(w^{(t)})$.

**Lemma 5** *Let $g^{(t)}$ be the unbiased inexact gradient aggregated by servers at time $t$. Under the assumptions of Theorem 2 we have*

$$\mathbf{E}\left[F(w^{(t+1)}) - F(w^{(t)})\right] \leq \left(L_{var} - \frac{M_t}{\gamma_t}\right) \left\|\Delta^{(t)}\right\|^2 + \left\|\Delta^{(t)}\right\| \left\|\nabla_{b_t} f(w^{(t)}) - \mathbf{E}\left[g^{(t)}\right]\right\| \tag{9}$$

*where the expectation is taken with respect to the random skip filter.*

**Proof.** For notation simplicity, we drop the index $t$ for the block indicator $b_t$, scaling matrix $U^{(t)}$, learning rate $\gamma_t$ and constant $M_t$ (recall that $M_t = \min_i U_i^{(t)}$ is the smallest coefficient-specific learning rate as induced by the Mahalanobis metric in the proximal operator).

First note that $g_b^{(t)} = g^{(t)}$ because the gradients are computed in block $b$. Hence it follows that also the update $\Delta^{(t)}$ is restricted to block $b$. By Lemma 4 we have that

$$\Delta_b^{(t)} = \text{Prox}_\gamma^U\left[w_b^{(t)} - \gamma U^{-1} g^{(t)}\right] - w_b^{(t)} = \Delta^{(t)}$$

and therefore $w_b^{(t+1)} = \mathrm{Prox}_\gamma^U(w_b^{(t)} - \gamma U^{-1}g^{(t)})$. Using Lemma 4 again, we have

$$\frac{U}{\gamma}\left(w_b^{(t)} - \gamma g^{(t)} - w_b^{(t+1)}\right) \in \partial h(w_b^{(t+1)})$$

Since $h$ is block separable we can decompose the updates to obtain

$$
\begin{aligned}
h(w^{(t+1)}) - h(w^{(t)}) &= h(w_b^{(t+1)}) - h(w_b^{(t)}) \\
&\leq \left\langle \frac{U}{\gamma}\left(w_b^{(t)} - \gamma U^{-1}g^{(t)} - w_b^{(t+1)}\right), w_b^{(t+1)} - w_b^{(t)} \right\rangle \\
&= -\frac{1}{\gamma}\left\|\Delta^{(t)}\right\|_U^2 - \left\langle g^{(t)}, \Delta^{(t)} \right\rangle \\
&\leq -\frac{M}{\gamma}\left\|\Delta^{(t)}\right\|^2 - \left\langle g^{(t)}, \Delta^{(t)} \right\rangle
\end{aligned}
\tag{10}
$$

On the other hand, only the entries of $w^{(t+1)}$ in block $b$ has been changed comparing to $w^{(t)}$, which satisfies the requirement of Assumption 1, therefore, by Lemma 3,

$$
\begin{aligned}
f(w^{(t+1)}) - f(w^{(t)}) &\leq \left\langle w^{(t+1)} - w^{(t)}, \sum_{i=1}^m \nabla_b f_i(w^{(t)}) \right\rangle + \sum_{i=1}^m L_{\mathrm{var},i}\left\|\Delta^{(t)}\right\|^2 \\
&= \left\langle \Delta^{(t)}, \nabla_b f(w^{(t)}) \right\rangle + L_{\mathrm{var}}\left\|\Delta^{(t)}\right\|^2
\end{aligned}
\tag{11}
$$

Combining (10) and (11), we have

$$
\begin{aligned}
\mathbf{E}\left[F(w^{(t+1)}) - F(w^{(t)})\right] &\leq \left(L_{\mathrm{var}} - \frac{M}{\gamma}\right)\left\|\Delta^{(t)}\right\|^2 + \mathbf{E}\left[\left\langle \Delta^{(t)}, \nabla_b f(w^{(t)}) - g^{(t)} \right\rangle\right] \\
&\leq \left(L_{\mathrm{var}} - \frac{M}{\gamma}\right)\left\|\Delta^{(t)}\right\|^2 + \left\|\Delta^{(t)}\right\|\left\|\nabla_b f(w^{(t)}) - \mathbf{E}\left[g^{(t)}\right]\right\|
\end{aligned}
$$

In other words, the amount of change between objective functions is bounded from above both by the amount of change in parameters $\Delta^{(t)}$ and by the discrepancy in the block gradient. ■

**Proof of Theorem 2.** We now have all ingredients to prove convergence to a stationary point. In a nutshell we must bound $\left\|\Delta^{(t)}\right\|$ and all else follows. Given time $t$, denote by the chosen block $b = b_t$. We first upper bound the term $\left\|\nabla_b f(w^{(t)}) - \mathbf{E}\left[g^{(t)}\right]\right\|$ in (9). By Assumption 1 we have for $1 \leq k \leq \tau$ that

$$\left\|\nabla_b f_i(w^{(t-k+1)}) - \nabla_b f_i(w^{(t-k)})\right\| \leq L_{\mathrm{cov},i}\left\|w^{(t-k+1)} - w^{(t-k)}\right\| = L_{\mathrm{cov},i}\left\|\Delta^{(t-k)}\right\|.$$

Due to the bounded delay, worker $i$'s model is only out of date at time $t$ in the range $t - \tau \leq t_i \leq t$. The *significantly modified* filter places an additional noise term $\sigma^{(t_i)}$ on the model. By design of the filter we use

$$\left\|\sigma^{(t_i)}\right\|_\infty \leq \delta_{t_i} = \mathcal{O}\left(\frac{1}{t_i}\right).$$

Futhermore by the random skip filter, the expectation of the unbiased inexact gradient aggregated at time $t$ is given by

$$\mathbf{E}\left[g^{(t)}\right] = \sum_{i=1}^m \nabla_b f_i(w^{(t_i)} + \sigma^{(t_i)}).$$

Then we have

$$\left\| \nabla_b f(w^{(t)}) - \mathbf{E}\left[g^{(t)}\right] \right\|$$

$$= \left\| \sum_{i=1}^{m} \sum_{k=1}^{t-t_i} \left( \nabla_b f_i(w^{(t-k+1)}) - \nabla_b f_i(w^{(t-k)}) \right) + \nabla_b f_i(w^{(t_i)}) - \nabla_b f_i(w^{(t_i)} + \sigma^{(t_i)}) \right\|$$

$$\leq \sum_{i=1}^{m} \sum_{k=1}^{t-t_i} \left\| \nabla_b f_i(w^{(t-k+1)}) - \nabla_b f_i(w^{(t-k)}) \right\| + \left\| \nabla_b f_i(w^{(t_i)}) - \nabla_b f_i(w^{(t_i)} + \sigma^{(t_i)}) \right\|$$

$$\leq \sum_{i=1}^{m} \sum_{k=1}^{t-t_i} L_{\text{cov},i} \left\| \Delta^{(t-k)} \right\| + L_{\text{cov},i} \left\| \sigma^{(t_i)} \right\|$$

$$\leq \sum_{i=1}^{m} \sum_{k=1}^{\tau} L_{\text{cov},i} \left\| \Delta^{(t-k)} \right\| + L_{\text{cov},i} \sqrt{p}\delta_{t-\tau}$$

$$= \sum_{k=1}^{\tau} L_{\text{cov}} \left\| \Delta^{(t-k)} \right\| + L_{\text{cov}} \sqrt{p}\delta_{t-\tau} \qquad (12)$$

where we used the fact that $\sigma^{(t_i)} = \sigma_{b_{t_i}}^{(t_i)}$ so that Assumption 1 can be applied and $\|x\| \leq \sqrt{p}\|x\|_\infty$. Substitute (12) into (9) in Lemma 5, we have

$$\mathbf{E}\left[F(w^{(t+1)}) - F(w^{(t)})\right] \leq \left( L_{\text{var}} - \frac{M_t}{\gamma_t} \right) \left\| \Delta^{(t)} \right\|^2 + \sum_{k=1}^{\tau} L_{\text{cov}} \left\| \Delta^{(t)} \right\| \left( \left\| \Delta^{(t-k)} \right\| + \sqrt{p}\delta_{t-\tau} \right)$$

$$\leq \left( L_{\text{var}} + \frac{L_{\text{cov}}\tau}{2} - \frac{M_t}{\gamma_t} \right) \left\| \Delta^{(t)} \right\|^2 + \sum_{k=1}^{\tau} \frac{L_{\text{cov}}}{2} \left\| \Delta^{(t-k)} \right\|^2 + L_{\text{cov}} p \delta_{t-\tau}^2$$

Summing over $t$ yields

$$\mathbf{E}\left[F(w^{(T+1)}) - F(w^{(1)})\right] \leq \sum_{t=1}^{T} \left( L_{\text{var}} + L_{\text{cov}}\tau - \frac{M_t}{\gamma_t} \right) \left\| \Delta^{(t)} \right\|^2 + L_{\text{cov}} p \delta_{t-\tau}^2 \qquad (13)$$

Denote by $c_t = \frac{M_t}{\gamma_t} - L_{\text{var}} - L_{\text{cov}}\tau$, since $\gamma_t \leq \frac{M_t}{L_{\text{var}} + L_{\text{cov}}\tau + \epsilon}$ for all $t$, then all $c_t \geq \epsilon > 0$. So

$$\epsilon \sum_{t=1}^{T} \left\| \Delta^{(t)} \right\|^2 \leq \sum_{t=0}^{T} c^{(t)} \left\| \Delta^{(t)} \right\|^2 \leq \mathbf{E}\left[F(w^{(1)}) - F(w^{(T+1)})\right] + L_{\text{cov}} p \delta_{t-\tau}^2 \qquad (14)$$

for any $T$. Since $\delta_t = O(\frac{1}{t})$, and by the fact that $1 + \frac{1}{2^2} + \frac{1}{3^2} + \ldots = \frac{\pi^2}{6}$. Then the RHS of (14) is constant when $T \to \infty$, which implies $\lim_{t\to\infty} \Delta^{(t)} \to 0$. So $\lim_{t\to\infty} \text{Prox}_{\gamma_t}^{U_t}(w^{(t)}) - w^{(t)} \to 0$, thus we find a local minimal point. ∎

## B  Sparse Logistic Regression

In addition to the largest experiment reported in Section 5, we present more experimental results on a range of sparse training data, as listed below. URL and KDDa are public sparse text datasets [2], while the click-through rate datasets CTRa and CTRb are subsampled from the dataset used in Section 5.

|                  | URL  | KDDa | CTRa | CTRb  |
|------------------|------|------|------|-------|
| # of examples    | 2M   | 8M   | 4M   | 0.34B |
| # of coordinates | 3M   | 20M  | 60M  | 2.2B  |
| # of nnz entries | 277M | 305M | 400M | 31B   |

We focus on the convergence of the objective value and runtime. More precisely, we report the relative objective values calculated via $\frac{F(w^{(t)})}{F(w^*)} - 1$ for each data pass, consisting of several iterations. An estimate of the optimal value $w^*$ is obtained by performing $4\times$ as many iterations as needed for convergence.

### B.1  Comparison to other algorithms

Since we were unable to identify distributed multi-machine sparse logistic regression algorithms capable of scaling to the dataset sizes in our research we compared to other solvers available for the *multicore* setting. This meant limiting ourselves to a relatively small dataset containing only millions of observations, as described in the table above.

More specifically, we compared our solver to shotgun[3] [7] on a single machine with 32 threads/workers. The results of CDN[4] (single thread shotgun) are also reported for reference. Figure 7 shows the objective values versus time. As can be seen, all three algorithms obtain similar objective values after 50 data passes, however, the parameter server is 4 times faster than both shotgun and CDN in terms of runtime.

The main reason can be found in the data partitioning strategies. Each thread of shotgun processes a single coordinate at a time, which often has irregular pattern of non-zero entries and therefore it is hard to perform load balancing. On the most high-dimensional dataset, CTRa, shotgun is even slower than the single thread version. On the other hand, the parameter server uses multi-thread linear algebra operators on a large block of the training data. This coarse-grained parallelization leads to better speedup.

### B.2  Scalability

We investigate scalability by increasing the number of workers from 16 to 256. The speedups of running times comparing to 16 workers are reported. A nine-fold speedup is observed when increasing the number of workers by 16 times.

## C  Reconstruction ICA

### C.1  Problem

Reconstruction ICA aims to find a sparse representation of the raw dataset. It relaxes Independent Component Analysis by allowing for an overcomplete solution [18]. Denote by $\{x_i\}_{i=1}^n \in \mathbb{R}^p$ the observations. The objective function of RICA has a nonconvex loss function $f(W)$ and convex but nonsmooth penalty $h(W)$.

$$\underset{W \in \mathbb{R}^{\ell \times p}}{\text{minimize}} \sum_{i=1}^{n} \frac{1}{2} \left\| WW^\top x_i - x_i \right\|_2^2 + \lambda \left\| W x_i \right\|_1, \tag{15}$$

Figure 7: Both shotgun and our algorithm use 32 threads on a single machine. Each point indicates one pass through the data. In total 50 passes are shown.

We denote by $X = (x_1, \ldots, x_n)^\top \in \mathbb{R}^{n \times p}$ the data matrix. The gradient of the smooth loss $f$ is

$$\nabla f(W) = W \left( (W^\top W - I) X^\top X + X^\top X (W^\top W - I) \right) \tag{16}$$

Figure 8: We tested scalability of the algorithm when increasing the number of clients (for a fixed number of servers) from 16 to 256. We achieve almost a perfect speedup. Much of the delay is likely due to the increased network load for the servers.

This can be seen by rewriting the objective function $f(W)$ by using a number of trace identities:

$$
\begin{aligned}
f(W) &= \frac{1}{2} \left\| XW^\top W - X \right\|_F^2 \\
&= \frac{1}{2} \operatorname{tr}\left( W^\top W X^\top X W^\top W - 2 X^\top X W^\top W + X^\top X \right) && (\|A\|_F^2 = \operatorname{tr}(A^\top A)) \\
&= \frac{1}{2} \operatorname{tr}\left( W^\top W X^\top X W^\top W \right) - \operatorname{tr} X^\top X W^\top W + \frac{1}{2} \operatorname{tr} X^\top X && (\operatorname{tr}(A+B) = \operatorname{tr}(A) + \operatorname{tr}(B)) \\
&= \frac{1}{2} \operatorname{tr}\left( X W^\top W W^\top W X^\top \right) - \operatorname{tr}\left( W X^\top X W^\top \right) + \frac{1}{2} \operatorname{tr}\left( X^\top X \right) && (\operatorname{tr}(AB) = \operatorname{tr}(BA))
\end{aligned}
$$

Applying (112) and (100) of [24] directly we obtain (16).

Unfortunately, invoking the proximal operator is nontrivial in the above case, since $\|WX\|_1$ has non-separable components (but it is block separable). This means that we need to solve each of the following $n$ independent optimization problems, e.g., using ADMM, simultaneously:

$$
\underset{u_i}{\text{minimize}} \ \frac{1}{2\gamma} \left\| u_i - z_i \right\|_{H_i}^2 + \lambda \left\| X u_i \right\|_1, \quad \text{for } i = 1 \dots n \tag{17}
$$

where $w_i \in \mathbb{R}^p$ denotes the $i$-th row of $W$ and we set $z_i = w_i - \gamma H_i^{-1} \nabla_i f(W)$. Here $\gamma$ is the learning rate and $H_i \in \mathbb{R}^{d \times d}$ is a scaling matrix to adjust the metric of the space. Following [10], we choose the scaling matrices by

$$
H_i(t+1)^2 = H_i(t)^2 + \operatorname{diag}(w_i^{(t)} - w_i(t-1))^2 \text{ for } t \geq 0 \quad \text{and} \quad H_i(0) = \mathbf{1}
$$

which can be computed locally. For convenience we drop the subscript $i$ from the proximal step of (17). Augmenting it with an auxiliary variable $y := Xu$ the augmented Lagrangian is given by

$$
L(u, y, \mu) = \frac{1}{2\gamma} \left\| u - z \right\|_H^2 + \lambda \left\| y \right\|_1 + \langle \mu, Xu - y \rangle + \frac{1}{2\theta} \left\| Xu - y \right\|^2. \tag{18}
$$

Correspondingly we obtain the update rules

$$
u \leftarrow \left( \gamma^{-1} H + \theta^{-1} X^\top X \right)^{-1} \left( \gamma^{-1} H z + X^\top \left( \theta^{-1} y - \mu \right) \right) \tag{19a}
$$

$$
y \leftarrow S_\lambda \left( \theta^{-1} Xu + \mu \right) \tag{19b}
$$

$$
\mu \leftarrow \mu + \theta^{-1} \left( Xu - y \right), \tag{19c}
$$

where $S_\lambda(\cdot)$ is the soft-thresholding function. Note that a worker can update its parameters independently if it has all observations and $W^\top W$, which is typically much smaller than $W$. Therefore we use partitioning by parameters for RICA. The server maintains $W^\top W$, while each worker has $X$ and a part of rows of $W$. In other words, the workers compute and retain parts of the parameter space.

## C.2 Experiment

Since most computations are dense matrix operations, we implemented the proposed algorithm on GPUs using CUBLAS. The latter uses all of the computational units within the GPU by default.

Figure 9: Reconstruct ICA on dataset ImageNet. Left: Varying delays on 16 GPU machines. Right: Decomposition of running times. Bottle: Scalability when increasing the number of workers from 1 to 16.

We run a single worker on each GPU card. We run the experiments on a cluster with each machine equipped with an Nvidia Tesla K20. The dataset used is ImageNet, which contains 100,000 randomly selected and resized $100 \times 100$ pixel images[5].

The experimental results are shown in Figure 9. Similar to $\ell_1$-regularized logistic regression, the clear improvement by asynchrony is also observed. Unlike the former, increasing the delay affects both runtime time and convergence only little. This is because the actual update delay of RICA is typically 1. When increasing the number of workers by 16 times, we see a 13.5 fold acceleration for RICA in Figure 9. The main reason why the speedup of RICA is better than $\ell_1$-regularized logistic regression is that RICA mainly consists of dense matrix operations. They are easier to balance than sparse matrices, and hence offer better scalability.

## Footnotes

[2]www.csie.ntu.edu.tw/~cjlin/libsvmtools/datasets

[3]www.select.cs.cmu.edu/projects/shotgun/

[4]www.csie.ntu.edu.tw/~cjlin/liblinear

[5]www.image-net.org