[Reviews · NeurIPS 2014]

Submitted by Assigned_Reviewer_32

This paper presents improvements on a system for large-scale learning known as "parameter server". The parameter server is designed to perform reliable distributed machine learning in large-scale industrial systems (1000's of nodes). The architecture is based on a bipartite graph composed by "servers" and "workers". Workers compute gradients based on subsets of the training instances, while servers aggregate the workers' gradients, update the shared parameter vector and redistribute it to the workers for the next iteration. The architecture is based on asynchronous communication and allows trading-off convergence speed and accuracy through a flexible consistency model. The optimization problem is solved with a modified proximal gradient method, in which only blocks of coordinates are updated at a time. Results are shown in an ad-click prediction dataset with O(10^11) instances as well as features. Results are presented both in terms of convergence time of the algorithm and average time spent per worker. Both are roughly half of the values for the previous version of the parameter server (version called "B" in the paper). Roughly 1h convergence time using 1000 machines each with 16 cores and 192Gb RAM, 10Gb Ethernet connection (800 workers and 200 servers). Other jobs were concurrently run in the cluster. The authors claim it was not possible to compare against other algorithms since at the scale they are operating there is no other open-source solution. In the supplementary material they do compare their system with shotgun [7] and obtain faster convergence (4x) and similar value of the objective function at convergence.

Since this is really a report of a working system, I have basically no technical comments to make. The numbers are impressive, the system works and there is no other open source system to compare against at this scale, so this is clearly a significant achievement. A nice materialization of a number of simple but important insights from the design of robust distributed systems into a machine learning pipeline (eg ideas like bounded delay, asynchronous push-pull, caching keys, etc.)

Novelty claimed with respect to "existing approaches":
- Solving large-scale problems (1000's nodes)
- Solving non-convex and non-smooth composite objective functions
Doesn't [10] addresses these aspects (apart from non-smoothness, not sure?) If you refer only to open source versions, then make this clear.

The aspect of the paper that I didn't like was the presentation.

The authors compare the results of their so-called "third-generation parameter server" to two other systems called "system A" and "system B" which are not cited and are presumably the previous versions of their system. One aspect of the paper that makes assessment harder is that the contributions of the paper are not very clear in a case in which this clarification is quite important. This is precisely because there have been previous versions of the system that have presumably not been published (otherwise systems A and B would have been cited, at least anonymously) and the current submission conflates novelties from the latest version with aspects that originates in the earlier systems. In my view it would have been better if either the paper was written as introducing the parameter server from scratch, and illustrating its evolution up to this point (if the goal were to sell the entire package) or instead that the *precise* differences between the third generation and the previous ones were made really clear in the paper (if the goal were to sell the 3rd generation, and not the entire package). From the presentation I believe the goal was the latter. The table in page 7 helps, but I was expecting a much more detailed discrimination.

Also, it's unfortunate that all the comparisons against other approaches (not the parameter server) are relegated to the supplementary material. I think this is a major weakness of the presentation.

My suggestion is that the authors improve the presentation of the paper. I suggest they tell the story as introducing the parameter server and describing 3 stages of its evolution. Also the paper would read better if the English was polished (although to me it's acceptable as it is since it doesn't compromise understanding).
Summary: The paper presents improvements on earlier versions of a system for large-scale linear learning (the parameter server). Results show roughly halving both the convergence time and average time per worker spent on computation when compared with the previous version of the parameter server. It's unfortunate that there are no references (not even anonymous) to either of the previous versions against which the proposed system is compared.

Submitted by Assigned_Reviewer_42

The paper describes an improved architecture for 'parameter server', a distributed learning framework that spreads both data and computation across the nodes, with a separate set of nodes dedicated to global state. An synchronous communication model that supports both push and pull for worker-server communication is used, and a bounded-delay ('flexible') consistency model that allows balancing between sequential and eventual consistency.

The paper attacks a general class of regularized optimization problems, focusing on the distributed subgradient algorithm that performs blockwise-updates using per-feature learning rates and lazy updates. The core innovation -- supporting the bounded delay consistency models -- allows support for the dependency graphs that arise naturally in algorithms for regularized optimization.

The KKT filter is clearly a central component to the design, as it is the primary throttling mechanism for feature selection. The specified procedure for estimation of global gradient is going to result in more aggressive filtration as the number of machines grows (unless \lambda is raised accordingly, which is not suggested in the paper). It appears that alternative methods (e.g., randomizing the described decision process) could be warranted. Additionally, it is unclear if "System A and System B" have equally aggressive feature selection. It would be informative to compare-contrast the results for different \lambda and demonstrate the effects of pruning.

The key piece missing from the paper is comparison of results to a fully lock-free variant of the system, such as one based on Hogwild. It is the most natural baseline architecture, as it skirts the consistency managements altogether. Additionally, results on standard public benchmarks would be much appreciated.

The paper appears to have been hastily put together, with numerous typos throughout.
Summary: The paper describes an iterative improvement for parameter server architecture, making it support varied-staleness consistency model, and evaluating on a block proximal gradient descent. Depth of experimental results can be improved significantly.

Submitted by Assigned_Reviewer_43

This paper describes a third-generation parameter server for distributed machine learning, which supports multiple consistency models. Then, the usefulness of the bounded delay consistency model, one of the models the framework provides, is justified by convergence guarantee of asynchronous proximal gradient algorithm the paper proposes. The paper also describes many implementation details such as user-defined filters and cacheing scheme, which should be very useful in massively large scale systems.

Quality: The proposed approach is strongly supported by empirical experiments which explores a variety of problems with (to my knowledge) the largest datasets machine learning community has ever seen. The paper also provides a theoretical convergence guarantee of the proposed algorithm.

Clarity: The paper is clearly written and easy to follow; the idea develops smoothly thanks to the presence of the motivating example, proximal gradient method. An unfortunate drawback of such a writing style is that the generality of the proposed framework is a bit lost. Clearly, the delayed proximal gradient algorithm should be just one of the examples the framework can provide; for example, the framework should also be able to support YahooLDA-style distributed Bayesian MCMC sampling [29] and delayed SGD [17]. It might be more desirable if authors could describe the class of machine learning algorithms well supported by the third generation parameter server; an abstract mathematical model for such algorithms, if possible, would be even better.

Originality: The idea of using parameter servers for distributed machine learning was already made popular thanks to [29] followed by [15], and this paper seems to present a refinement of the idea rather than a fundamentally different approach. Theorem 2 is a nice contribution, but it is not a very surprising result considering similar results on other algorithms such as [7,17, 26] already exist.

Significance: The parameter-server approach for distributed machine learning is one of the most promising approaches to scale machine learning for industrial-scale data, and I believe this paper provides a significant contribution by addressing practical concerns that can only arise in such a scale and not yet widely-known in the machine learning community.
Summary: This paper addresses important practical concerns that can arise when applying parameter server for peta-scale machine learning problems, and is strengthened by a useful convergence theorem of an algorithm it supports.
Author Feedback
Author rebuttal: * Reviewer 32:

Q. Why is it called a “the third-generation parameter server” and what is the novelty with respect to “existing approaches”, e.g. [10].

The first-generation of parameter servers was introduced in [2,29]. It was improved by special purpose systems such as Google DistBelief [10] and YahooLDA [2]. The proposed system, which we refer to as a 3rd generation system, is general purpose. It solves a superset of problems and scales to 100x to 1000x larger datasets.

The novelty lies in the codesign and synergy achieved by picking the right systems techniques, adapting them to ML algorithms, and modifying the ML algorithms to be more systems-friendly. This results in the first general purpose ML system demonstrated to scale to industrial data. From the systems aspect, specific contributions include an adaptive dependency graph, a (key,value) storage with smart updates on the master, hot failover for inference systems, and message filters. From the machine learning aspect, we propose a system-friendly variant of the proximal gradient algorithm which is able to solve nonconex and nonsmooth problems, and also prove the convergence under the relaxed consistency model.

Q. Details about System-A and B.

Both systems are commercial tools contributing more than 90% revenue of a multi-billion internet company. They adopted the parameter server architecture to solve sparse logistic regression. The differences to the proposed system are summarized on page 7. Implementation specific differences are that Systems A and B use C and proprietary libraries, whereas our system is implemented in C++ using open source code.

The reason for comparing to these two systems is: 1) there is no open source system scaling to the problem scale we evaluated; 2) System A and B are well optimized to form a challenging baseline. In particular, System-B is developed by one of the authors of this paper with his team during more than two years.

* Reviewer 42:

Q. The KKT filter:

Q.1 The procedure for estimating the global gradient will result in more aggressive filtration as the number of machines grows.

There are two solutions to this problem. One is changing the filter threshold. Recall that a feature is filtered if |estimated_gradident| < lambda - delta, where lambda is the coefficient of the l1 regularizer. Increasing delta can reduce the aggressiveness of the filter.

Another solution is to ask the server nodes to determine whether or not a feature will be filtered. In other word, at time t the filtering is determined by the aggregated global gradient at time t-1. Therefore this filtering procedure is not affected by the number of machines. But we also make an approximation here, the threshold delta is still necessary for tuning the aggressiveness.

Furthermore, we set delta = eta * max_optimal_violation, where eta is a parameter specified by hand while the later measure the convergence status. Tuning eta is less sensitive than tuning delta directly. In addition, we stop the filtering periodically to get back features mistakenly filtered earlier.

We evaluated all these heuristic ways for filtering features, but found their difference not to be significant on our dataset, especially after using the periodical stopping trick. In other words, the difference of number of nonzero weight in the final model is less than 10%, and the difference of test AUC is less than 0.05%.

Q.2 Do System A and B use KKT filter?

System B does. It is similar to the second approach mentioned in Q.1 but with a different rule of updating delta, which results in a slightly less aggressive filter comparing to the one used by the proposed parameter server.

Q.3 Compare the results of lambda, namely the coefficient of the L1 regularizer.

We tested lambda=1, 4, and 10. The system comparison results are very similar. Since lambda=4 gives better test accuracy we only reported its results.

(PS. If the reviewer’s intention is on how delta affect the system, then we answered in Q.1)

Q. Missing comparison to fully lock-free system such as Hogwild.

Hogwild is very different: Firstly, we are not aware of a working method to solve the nonsmooth L1 regularized logistic regression via Hogwild. Secondly, Hogwild is designed for shared memory architecture of 10s of cores in a single machine. We scale to 1000s of computers and 10,000s of cores.

We compared with another popular multithreaded algorithm, Shotgun, in Appendix C. The Parameter server outperforms Shotgun because of its near lock-free implementation: locking occurs only when waiting for the bounded-delay consistency model. If we set the maximal allowed delay to infinity, then we have a pure lock-free implementation. However, it is not desirable under the multiple machines setting. See Figure 6 in Appendix C for example, a too large delay could significantly affects the convergence rate.

Q. Results on standard public benchmarks.

In Appendix C, we compared Shotgun on public text datasets such as URL and KDDa. In Appendix D, we tested reconstruction ICA on the popular image dataset ImageNet.

* Reviewer 43:

Q. The paper did not discuss how other algorithms fit into the proposed system.

In this paper, we used the delayed proximal gradient algorithm as the motivation example to address the design decisions of the parameter server and evaluate the system performance. But as mentioned by the reviewer, other algorithms such as the MCMC sampling used by YahooLDA and delayed SGD used by DistBelief fit into the proposed system naturally. Due to the space constraint, these materials are relegated to other manuscripts such as [4].

Q. An abstract mathematical model of algorithms supported by parameter server.

This is nontrivial even for much simpler paradigms such as MapReduce which differs from our approach by being a directed bipartite graph whereas our approach is a bidirectional bipartite graph.